# Dynamic Modeling and Chaos Control of Informatization Development in Manufacturing Enterprises

**DOI:** 10.3390/e23060681

**Published:** 2021-05-28

**Authors:** Peng Niu, Jianhua Zhu, Yanming Sun

**Affiliations:** 1School of Business Administration, South China University of Technology, Guangzhou 510641, China; bmniupeng@mail.scut.edu.cn; 2School of Aviation Flight Basics, Naval Aviation University, Yantai 264001, China; 3School of Business Administration, University of Guangzhou, Guangzhou 510006, China; sunyanming@gzhu.edu.cn

**Keywords:** top management support, technical ability of employees, informatization performance, chaos recognition, chaos control

## Abstract

To explore the cooperative evolutionary mechanism among top management support, employees’ technical ability, and informatization performance in the process of the “integration of informatization and industrialization (IOII)” in manufacturing enterprises, this study established a three-dimensional dynamic model of informatization development, obtained the model parameters by the expert scoring method of case companies, and analyzed the time series of the dynamic model. After adjusting those parameters of the evolutionary process that do not meet the expectations of the enterprise, combined with management practice, the dynamic system is finally stable at the expected value. For a special state in the evolutionary process, the maximum Lyapunov exponent is used to identify the chaotic characteristics of the system, and a linear controller is designed to manage and control the chaotic system so that it evolves toward the expected value. The results of the case analysis verify the rationality of the model and the effectiveness of the control method, reveal the internal evolutionary mechanism of the informatization development of manufacturing enterprises, and explain the influence of chaos on enterprise management so as to help managers to use and control chaos.

## 1. Introduction

“Industry 4.0” is driving the development of intelligent manufacturing models. Intelligent manufacturing, equipped with information technology, sensors, and advanced equipment, involves a high degree of collaboration between the industrial chain and the service chain, and can work independently and by interacting with the production environment. The core of the “Made in China” brand is information technology. The integration of industrialization and informatization applies information technology to all aspects of industrial production, changes the traditional industrial production method, and uses informatization as a conventional means of industrial enterprise management. With the emergence and rapid development of emerging technologies, such as the Internet of Things [1], cloud computing [2], industrial big data [3], and “5G + Industrial Internet” [4], the application of information technology in manufacturing companies is bringing new ideas to the forefront. At the same time, this has made the integration of a new generation of information technology and manufacturing technology more complicated.

Manufacturing informatization is a system engineering method that is subordinate to a complex system. Informatization can help enterprises improve operations, management, product development, the production of manufacturing enterprises, production efficiency, product quality, and corporate innovation capabilities; reduce consumption; and drive innovation in product design methods and design tools, thereby comprehensively enhancing the competitiveness of enterprises. In recent years, with the increase in the complexity of enterprises’ informatization construction, scholars have gradually begun to use the method of complex systems to study the informatization of manufacturing enterprises. Zhu et al. [5] regarded the integration of manufacturing technology and information technology as a complex system, and then used a synergetic method to study the integration process, revealing the dynamics of technology fusion. Later, Zhu et al. [6] used technical efficiency as an indicator to measure the integration of informatization and industrialization and used self-organization and convergence theories to discuss the framework for the internal dynamic evolution of manufacturing technology and information technology. On this basis, Zhu et al. [7] modeled and analyzed the three dimensions of technology integration, product integration, and business integration from the perspective of quality management. The research results pointed out that the process of informatization is a dynamic process, and it is inevitable to encounter the effects of chaos. According to the statistics of China’s integration of informatization and industrialization platforms, as of the beginning of 2019, the total number of enterprises participating in informatization and industrialization reached 1,419,571, but only 7.7% of them have realized intelligent manufacturing. The shortage of professional talent and the informatization strategy of corporate managers are the main reason why Chinese manufacturing is “big but not strong.” Therefore, it is important for Chinese manufacturing enterprises to upgrade to intelligent systems, and to grasp the internal relationship between information performance and employee skills and high-level support in the process of in-depth integration.

Scholars’ research on manufacturing informatization performance mostly uses empirical methods to explore the relationship between informatization input and performance, the impact of informatization input on organizational performance and cost management, and the evaluation method and index system of informatization performance. Yu et al. [8] established the demand function of IT investment and used empirical research methods to show the relationship between IT investment and different types of diversified companies and concluded that information technology can serve as an effective coordination and control mechanism for moderate diversification. The coordination and control mechanism of non-IT organizations is more appropriate in the context of a higher degree of diversification. Li et al. [9] measured informatization from the three aspects of IT managerial capability, IT-enabled capability, and IT infrastructure capability. Through empirical research, they found that senior executive support has a significant impact on IT-enabled capability, IT managerial capability and IT, but has no significant relationship with infrastructure capability. Niu [10] used empirical methods to verify that service-oriented manufacturing information system customization has a significantly positive impact on its performance, and data flexibility, process flexibility, and system flexibility play a chain-like intermediary role between them. Noor et al. [11] studied the relationship between IS complexity and the performance evaluation system. Through a component-based structural equation, the research concluded that the maturity of the information system was the determinant of the performance measurement standard, and the different dimensions of IS complexity. Shiau et al. [12] explained from the perspective of information transmission that the investment in the construction of enterprise information equipment and the establishment of the website can promote the sharing of demand, inventory, output, and other information with upstream and downstream enterprises in the supply chain, and through this valuable information of sharing, promotes corporate performance. Li et al. [13] believe that information technology investment and changes in inventory standards can significantly improve the inventory performance of manufacturing companies. Bayraktar et al. [14] conducted an empirical study on a sample of 203 manufacturing companies in Turkey and demonstrated that supply chain management and information system practices have a positive impact on business performance. Strecker [15] proposed a performance evaluation index system for manufacturing informatization from the impact of informatization on enterprise operating costs, efficiency, quality, and management. Sircar et al. [16] proposed an analysis framework for the relationship between corporate performance and IT and corporate investment.

Scholars have mostly studied the impact of senior support and employee skills on informatization performance separately; few scholars have studied the two impacts simultaneously. Lin et al. [17] used empirical methods from the three aspects of information technology, organization, and the environment to discuss how the support of high-level informatization will be affected by factors such as technical superiority, technical complexity, organizational readiness, and organizational information density. Environmental competitiveness is a moderating variable. Papke-Shields et al. [18] believe that the top management support of manufacturing enterprises indirectly affects the economic performance of enterprises through environmental innovation. Chao et al. [9] found a significantly positive correlation between executive support and information technology capabilities. However, no significantly positive correlation was found between senior management support, information technology management and infrastructure capabilities. Information technology management capabilities, information technology support capabilities, and information technology infrastructure capabilities have a significantly positive impact on breakthrough innovation performance. Burrows et al. [19] constructed a model of the relationship between executive support and informatization performance, explaining that executive beliefs and participation in informatization construction are beneficial to information system management and control, informatization strategy, and information system and business matching, thereby improving enterprise information leadership and promoting the success of information systems. Shu et al. [20] believe that top management support is an important driving factor in the process of knowledge sharing. Leten et al. [21] found that there is a significantly positive correlation between employees’ general skills and product innovation performance, and there is a significantly positive relationship between employees’ specific skills and process innovation performance. Innovation strategy has a significantly positive correlation with employee skills and organizational performance. Djoko et al. [22] used a multiple linear regression model to analyze 125 randomly sampled service industry employee data points and found that information technology products, knowledge, and skills have a significant impact on employee performance.

Through the abovementioned literature review, we found that most scholars now use empirical methods to study informatization construction. However, as mentioned above, the informatization construction process of manufacturing enterprises is a complex system, and the empirical research method is result-oriented, so it cannot reveal the dynamic process and internal evolutionary mechanism in the informatization construction. Although some scholars have conducted research on the process of informatization construction, they are all based on technology integration, product integration, and business integration aspects. They rarely study the relationship and co-evolutionary mechanism between high-level support, employee technical capabilities, and informatization performance in the process of the integration of informatization and industrialization in manufacturing enterprises. As mentioned above, the biggest difficulty in the process of informatization construction is the shortage of professional talent and the lack of a clear understanding of informatization strategy by corporate managers. Therefore, this study attempts to establish a dynamic evolution model, analyzing the evolutionary trajectory with a case data analysis and controlling any evolutionary processes that do not meet the expectations of the enterprise. This paper provides guidance for the transformation and upgrading of manufacturing enterprises.

## 2. Manufacturing Enterprise Informatization Development Dynamic Model

Combined with the literature and the research data of enterprises in the Pearl River Delta region of China, this study defines top management support as vision propaganda, resource support (financial and human resources), reform support, consultant support, and active participation. This is mainly reflected in the company’s vision of transformation and upgrading from industrialization to informatization; doing a good job of mission propaganda, according to the blueprint of enterprise development; investing and managing the software and hardware of informatization projects; introducing high-tech talent; providing consultation and coordination for informatization construction; promoting technological reform and innovation; encouraging the learning of new technologies and knowledge from benchmark enterprises in the same industry, industry forum summits and experts; leading the development directions of large-scale project plans and project portfolios and managing them; actively participating in the implementation of large-scale projects; improving the staff training system; encouraging employees to participate in the assessment of international general qualification certificates; improving performance appraisal and promotion channels; and creating an organizational environment conducive to the development of information technology.

The technical ability of employees includes the basic quality, general skills, and professional skills of employees in an organization. The basic qualities include education level, working years, age, and position. General skills include expression ability, communication ability, adaptability, organization ability, teamwork ability, stress tolerance, acceptance of new technology, and knowledge and innovation ability. Professional skills include information processing ability, problem-solving ability, and professional information technology ability.

Informatization performance is the quality of the information system itself and the performance of applying information technology. The quality of an information system includes the authenticity, integrity, and timeliness of the data; whether different systems have unified basic data; whether various information systems are interconnected; whether different organizations and departments can cooperate according to business needs; and whether it can meet the needs of a business’s processes and workflow. The performance of applying information technology includes the following: degree of cost saving of a digital office; how to improve work performance and the comprehensive competitiveness of enterprises; whether to improve the quality of products and services; the degree to which the management methods and ideas of information systems are accepted by the organization, such as the working methods and performance appraisal methods of procurement, production, sales, and service; the degree of information system-aided decision making; and the impact on the satisfaction degree of upstream and downstream stakeholders (including suppliers, employees, and customers) in the supply chain.

The conceptual structure is shown in Table 1.

Let the top management support be *x*, the technical ability of employees be *y*, and the informatization performance be *z*, and we can establish a dynamic evolution model of the informatization development of manufacturing enterprises as follows:(1)$$\left\{\begin{array}{l} \frac{dx}{dt}=a_{1}x-a_{2}z\\ \frac{dy}{dt}=-b_{1}xy+b_{2}y+b_{3}x^{2}\\ \frac{dz}{dt}=c_{1}x-c_{2}y+c_{3}xy-c_{4}z \end{array}\right.,$$
where $a_{1}$ represents the self-influence factor and $a_{2}$ indicates the impact of informatization performance on top management support. With the improvement of the informatization level, top management will reduce financial and human resources investment in informatization construction. $b_{1}$ indicates that the synergy between top management support and employees’ technical ability will affect employees’ technical ability. $b_{2}$ indicates that the current technical ability of employees promotes the improvement of the future technical level, and the demand of enterprises for technical ability in employees urges employees to improve their technical ability. $b_{3}$ indicates that the top management pays attention to training, and a good organizational environment will greatly improve the technical ability of employees. $c_{1}$ indicates that top management investment and attention to informatization construction can promote informatization performance. $c_{2}$ indicates that the current technical ability of employees will inhibit informatization performance. $c_{3}$ indicates that the synergy between top management support and employees’ technical ability will effectively improve the informatization performance. $c_{4}$ indicates that the current informatization status of enterprises will inhibit the future informatization performance.

## 3. Case Analysis

### 3.1. Mathematical Model and Co-Evolution Analysis

Taking the Guangzhou HC equipment manufacturing enterprise as the research object, in accordance with the contents of Table 1, the informatization investigation was carried out in three factories. The company had been a traditional operation for a long time. Under the background of China’s vigorous promotion of IOII, factory 1 took the lead in carrying out informatization reform. By the end of the investigation in 2020, it had basically realized the digitalization of industrial e-commerce and key processes. The informatization levels of factories 2 and 3 were relatively low. Although it started slowly, the head office has a very clear plan for factory 2 and hopes to introduce advanced equipment, such as mechanical arms, to realize production automation, flexible manufacturing, and more diversified and customized products. There are many product lines in factory 3, but most of them are not the main business, with fewer sales and a relatively older average age of the employees, so the informatization reform is difficult and progress is slow. According to the characteristics of these three factories, this study used the expert scoring method to score the parameters on a scale of 0 to 10 (where 0 is very low and 10 is very high). To ensure the reliability of the data, the expert consultation form was filled in anonymously. Let parameters $\alpha =(a_{2},b_{1},b_{3},c_{1},c_{2},c_{3}),\beta =(a_{1},b_{2},c_{4})$, among which *α* are filled by 10 people from the top management of the head office, factory managers and department heads, and *β* are filled by six people from the research group experts. According to the results of consultation, the parameter values are calculated $\alpha_{i}=\sum_{i=1}^{10}\alpha_{i}\omega_{i},\beta_{i}=\sum_{i=1}^{6}\beta_{i}\omega_{i}$, the weights *ω* are set by the top management of the head office and experts of the research group. According to the weights and scores of experts in Table A1 in Appendix A, the system parameters of the informatization construction of the three factory areas of the company are obtained as follows:$$\begin{array}{l} (2.48,0.27,3.05,0.44,8.38,0.54,2.43,6.85,4.09)\\ (3.07,6.75,8.58,4.03,7.42,8.61,5.04,8.17,3.71)\\ (4.02,0.34,3.13,0.19,4.00,0.22,2.15,7.05,3.89) \end{array}$$

Substituting the three group parameters into the dynamic system (1), we can obtain the evolutionary model of the dynamic system of informatization promotion for each factory:(2)$$\mathrm{Factory}\;1:\;\left\{\begin{array}{l} \frac{dx}{dt}=2.48x-0.27z\\ \frac{dy}{dt}=-3.05xy+0.44y+8.38x^{2}\\ \frac{dz}{dt}=0.54x-2.43y+6.85xy-4.09z \end{array}\right.$$
(3)$$\mathrm{Factory}\;2:\;\left\{\begin{array}{l} \frac{dx}{dt}=3.07x-6.75z\\ \frac{dy}{dt}=-8.58xy+4.03y+7.42x^{2}\\ \frac{dz}{dt}=8.61x-5.04y+8.17xy-3.71z \end{array}\right.$$
(4)$$\mathrm{Factory}\;3:\;\left\{\begin{array}{l} \frac{dx}{dt}=4.02x-0.34z\\ \frac{dy}{dt}=-3.13xy+0.19y+4.00x^{2}\\ \frac{dz}{dt}=0.22x-2.15y+7.05xy-3.89z \end{array}\right.$$

Next, we analyze the evolution of the state variables of the above dynamic system over time, and then discuss the future development trajectory of informatization construction in each factory. At first, systems (2) and (3) are simulated; the initial values are (2, 5, 2) and the duration is 200. The time series diagram and phase diagram, evolving over time, are shown in Figure 1, Figure 2 and Figure 3.

As shown in Figure 1, the informatization development of factory 1 is stable. After a period of informatization construction, it gradually stabilizes at (2.4385, 6.3259, 22.3977), that is, when the top management support of factory 1 is 2.4 and the technical ability level of the employees is 6.3, the company can achieve an informatization performance score of 22.4.

As shown in Figure 2 and Figure 3, the informatization development of factory 2 is stable. After a period of informatization construction, it gradually stabilizes at (0.5241, 4.6020, 0.2233), that is, when the top management support of factory 2 is 0.5 and the technical ability level of employees is 4.6, the company’s information performance score is only 0.2. In addition, from Figure 3, we can see that the top management support and technical capabilities of the employees in factory 2 during the informatization construction have great fluctuations in terms of informatization performance, indicating that the informatization construction of factory 3 lacks scientific guidance. By comparing Figure 1 and Figure 2, we see that factory 1 receives more support from senior managers, and its employees have stronger technical capabilities, so factory 1′s informatization performance is better.

From Figure 1 and Figure 2, we see that when the evolution of systems (2) and (3) converges to a point, the informatization transformation and upgrading of factories 1 and 2 have reached a stable state. If there is no subsequent innovative strategic deployment, the informatization construction of the two factors will stagnate at the point of equilibrium.

Obviously, the evolution of the two systems converges at a particular point, which shows that the informatization development strategy of the head office for the two factories has made the transformation and upgraded from industrialization to informatization in the factory, reaching a stable state. If there is no innovative strategic deployment in the follow-up, the informatization construction of the two factories will stagnate at the equilibrium point. In the initial stage of dynamic system (2), by improving employees’ technical ability, introducing high-tech talents, strengthening business training, increasing investment in information software and hardware, and inviting industry experts to give lectures, all kinds of behaviors promote the remarkable improvement of information performance. Then, based on ensuring a high level of information performance, the top management support is slowly reduced, making the dynamic system finally stable at the equilibrium point (2.4385, 6.3259, 22.3977). However, the dynamic system (3) lacks the support of senior managers, which means the informatization construction of factory 2 lacks investment. The development of informatization stabilizes at the equilibrium point (0.5241, 4.6020, 0.2233) after a period of fluctuations.

According to the actual operating conditions of factory 2, the equilibrium point of dynamic system (3) is not the best practice. This is because in the early stage of informatization construction, the person in charge of Party A changed during the implementation of the two major information system projects; the project handover was not timely, and the manager of the technical center of the factory did not grasp the key nodes of project management, which led to the slow introduction of information equipment. The information systems could not be fully interconnected and could not fully match the current business process. Even though the employees had a strong working and innovative ability, the informatization performance was still low. It was decided by the managers of the head office through consultation that the factory would realize flexible manufacturing and a service-oriented production management process in the future. The top management should play the role of chief planner and organizer, increase the input of equipment and personnel, and encourage employees to learn from benchmarking enterprises; the factory manager must provide strong support, strengthen project management, and accelerate the progress of informatization construction to meet future strategic needs. Therefore, the influence coefficient of informatization performance on top management support should be reduced—that is, adjust a_2_ to 0.39 so that the new dynamic system can realize the increase of top management support in the early stages, promote the remarkable improvement of informatization performance, ensure the reasonable development of the three aspects in the later stage, and finally converge to the equilibrium point (3.0884, 3.1494, 24.3118). With the same initial value, a new three-dimensional phase diagram of dynamic system evolution, meeting the needs of enterprises, is shown in Figure 4.

For the convenience of comparison, next, under the same initial conditions, the evolutionary trajectory of the dynamic system (4) is shown in Figure 5 and Figure 6.

It can be seen that system (4) does not evolve toward a certain stable point. Due to competition and cooperation among top management support, employees’ technical ability, and information performance, their values show irregular and quasi-random movement patterns in a certain range with the passage of time, which indicates an unstable and nonconvergent evolutionary process. This means that the head office’s informatization strategic deployment of factory 3 makes the informatization performance go up and down, and sometimes even hinders the development of business processes. The top management cannot provide reasonable support according to the current informatization performance, and so the staff are not well trained, and their skills do not steadily improve. The slight changes in the top management support and staff technical skills will cause exponential changes in informatization performance. The slight changes in the initial values of dynamic systems will cause huge differences in their future state after repeated iterations, resulting in a completely different evolution. This phenomenon of “a tiny error may lead to a large discrepancy” is obviously an internal randomness and interaction mechanism within the deterministic system. In fact, due to its long-term reliance on manual work, this factory has a low level of informatization, an older average age of its employees, and uneven technical skills. It lacks practical experience in the early stage of informatization construction and does not accurately grasp the important strategic position of informatization in the process of enterprise development, so cannot manage the factory efficiently and dynamically, thus falling into a vicious cycle. This uncertain nonperiodic and nonconvergent evolutionary process is caused by a deterministic system and is sensitive to the initial value, basically conforming to the characteristics of chaos. Then, the chaotic characteristics are identified to determine whether the dynamic system is a chaotic system.

### 3.2. Identification of Chaotic Characteristics

Chaos is widely used in research in the natural and social sciences, and many scholars combine chaos theory with manufacturing industry and industrial engineering [23,24,25]. In research on chaos, the Lyapunov exponent is usually used to measure the average exponential rate of convergence or divergence in a phase trajectory over time [26,27]. When the maximum Lyapunov exponent is less than 0, it shows that the system is stable and the trajectory will tend to a certain stable point [25,28,29]. When the maximum Lyapunov exponent is equal to 0, it shows that the system is unstable, and the trajectory may be periodic or bifurcated. When the maximum Lyapunov exponent is greater than 0, the system will reach a chaotic state—that is, no matter how small the initial distance of the phase trajectory, the difference will increase exponentially with the passage of time and cannot ultimately be predicted. Therefore, with the same initial value, the difference between the two initial points being 10^−7^, and considering the synergistic effect of top management support and employees’ technical ability on informatization performance, the maximum Lyapunov exponent of system (4) about parameter *c_3_* is calculated by the classical Benettin calculation method.

The chaotic identification steps of system (4) are as follows:Step 1: Take a minimum value $d_{0}=10^{-7}$; the range of *c*_3_ is [6,8],
$$a_{1}=4.02{,a}_{2}=-0.34{,b}_{1}=-3.13{,b}_{2}=0.19{,b}_{3}{=4,c}_{1}=0.22{,c}_{2}=-2.15{,c}_{4}=-3.89$$Step 2: With the initial values $X_{1}=\left(2,5,2\right)$ and $X_{2}=\left(2,5,2+d_{0}\right)$, and time $t_{0}=\left(0,500\right)$, substituting $X_{1},X_{2}$ into Equation (4), we obtain $Y_{1}=f(X_{1})$ and $Y_{2}=f(X_{2})$.Step 3: Make the last line of $Y_{1}$ and $Y_{2}$ be $\left(x_{1},y_{1},z_{1}\right)$ and $\left(x_{2},y_{2},z_{2}\right)$, calculate the distance $d_{1}=\sqrt{{\left(x_{1}-x_{2}\right)}^{2}+{\left(y_{1}-y_{2}\right)}^{2}+{\left(z_{1}-z_{2}\right)}^{2}}$ between two points, and then make the coordinates of the next point in phase trajectory be the following:$$X_{i+1}=(x_{i}+\frac{d_{0}}{d_{1}}(x_{i+1}-x_{i}),y_{i}+\frac{d_{0}}{d_{1}}(y_{i+1}-y_{i}),z_{i}+\frac{d_{0}}{d_{1}}(z_{i+1}-z_{i}))$$Step 4: Repeat steps 2 and 3; when i>50, calculate $lya_{i-50}=\frac{1}{i-50}\sum_{i=50}^{100}\log\frac{d_{1}}{d_{0}}$.Step 5: Repeat steps 2–4 to obtain each *lya* corresponding *c*_3_.

The curve of the maximum Lyapunov exponent changing with parameter *c*_3_ is shown in Figure 7.

When *c*_3_ = 7.05 (Figure 7), the maximum Lyapunov exponent of the dynamic system (4) is 0.384, indicating that the informatization construction of factory 3 was already in a chaotic state. In other words, with the passage of time, the top management cannot provide reasonable support according to the current informatization level, and the skills of employees cannot be improved steadily. The informatization performance will “get half the results with double the effort.” After repeated iterations, there will be exponential differences between the three values, which makes the informatization construction deadlocked and unpredictable, and even causes it to fail in serious cases. Obviously, the chaos state does not meet the expectations of the enterprise. Although factory 3 has its own problems, such as the late start, difficult promotion, and slow development of information construction, the product structure is mostly non-main products, and the skills of its employees are relatively weak. The head office hopes that the research team can give reasonable recommendations to help the factory embark on transformation and upgrading, so it is necessary to control the chaos of the dynamic system (4).

### 3.3. Chaotic Linear Control

Whether the chaos is good or bad, it should be controlled in combination with enterprise development planning. The sensitivity of a chaotic system to the initial value may not be a bad thing. As long as managers ensure the iteration results that they want and exert control over a chaotic system, evolving into a chaotic state can represent a new opportunity for enterprises. The purpose of chaos control is to eliminate chaos, break the chaotic state, re-enter a new equilibrium state, maintain stable development in the new informatization construction, achieve higher informatization performance, and rationally adjust the relationship between top management support and employees’ technical ability. Since chaotic linear control is easy to create, understand, and calculate [30,31], this study designs a linear controller:

Considering $f(X)=\frac{dX}{dt}=\left\{\begin{array}{l} \frac{dx}{dt}\\ \frac{dy}{dt}\\ \frac{dz}{dt} \end{array}\right.,x,y,z\in R$, linear controller μ=AX+B, in which $A=-J_{f}(X^{*})+C$, $J_{f}(X^{*})$ is the Jacobi matrix of f(X) at the expected value $X^{*}$, and *C* is the diagonal matrix, $B=-f(X^{*})-AX^{*}$.

Then the controlled system g(X)=f(X)+μ will converge to the expected value $X^{*}$.

**Proof : ** $$\begin{array}{l} g(X)=f(X)+\mu \\ =f(X)+(-J_{f}(X*)+C)X-f(X*)-(-J_{f}(X*)+C)X*\\ =f(X)-f(X*)+(-J_{f}(X*)+C)(X-X*) \end{array}$$
□

Jacobi matrix $J_{g}(X*)=J_{f}(X*)-J_{f}(X*)+C=C=\left[\begin{array}{l} \lambda_{1}\;\;\\ \;\lambda_{2}\;\\ \;\;\lambda_{3} \end{array}\right]$ of g(X) at point $X^{*}$, and then, the eigenvalue of $J_{g}(X*)$ is $\lambda_{1},\lambda_{2},\lambda_{3}$.

Because $\frac{\partial g}{\partial X}\left|{}_{X=X*}\right.=\underset{X\to X*}{\lim}\frac{g(X)-g(X*)}{X-X*}=C$, $\lambda_{1},\lambda_{2},\lambda_{3}$ can also represent the segregation index of g(X).

When $\lambda_{1}<0,\lambda_{2}<0,\lambda_{3}<0$, the eigenvalue of $J_{g}(X*)$ of g(X) at point $X^{*}$ is less than 0, so g(X) is convergent, and since $g(X^{*})=f(X^{*})-f(X^{*})=0$, point $X^{*}$ is the asymptotic stable point of g(X).

Because factory 3 wants the dynamic system to be stable at the expected value of $X^{*}={\left(3,8,30\right)}^{T}$, the evolutionary trajectory of the system after control should be convergent—that is, the system segregation index is negative. Here, $\lambda_{1}=-0.3,\lambda_{2}=-0.5,\lambda_{3}=-0.5$, so system (4) is linearly controlled:$$J_{f}(X^{*})=\left[\begin{array}{ccc} 4.02 & 0 & -0.34\\ -1.04 & -9.2 & 0\\ 56.62 & 19 & -3.89 \end{array}\right],\;C=\left[\begin{array}{lll} -0.3 & & \\ & -0.5 & \\ & & -0.5 \end{array}\right]$$

Therefore,
$$A=\left[\begin{array}{ccc} -4.32 & 0 & 0.34\\ 1.04 & 8.7 & 0\\ -56.62 & -19 & 3.39 \end{array}\right],\;B=-\left[\begin{array}{c} 1.86\;\\ -37.6\\ 35.96 \end{array}\right]-\left[\begin{array}{c} -2.76\;\\ 72.72\\ -220.16 \end{array}\right]=\left[\begin{array}{c} 0.9\;\\ -35.12\\ 184.2 \end{array}\right]$$

Linear controller
$$\mu =\left\{\begin{array}{l} -4.32x+0.34z+0.9\\ 1.04x+8.7y-35.12\\ -56.62x-19y+3.39z+184.2 \end{array}\right.$$

Substitute system (4) to obtain the controlled system as follows:(5)$$g(X)=\left\{\begin{array}{l} \frac{dx}{dt}=-0.3x+0.9\\ \frac{dy}{dt}=1.04x-3.13xy+8.89y+4.00x^{2}-35.12\\ \frac{dz}{dt}=-56.4x-21.15y+7.05xy-0.5z+184.2 \end{array}\right.$$

Then, under the same initial conditions, the evolutionary trajectory of dynamic system (5), as shown in Figure 8, is used to verify the effectiveness of the designed linear controller.

From Figure 8, we see that, after applying the controller *μ*, system (5) is no longer in a chaotic state but enters a new equilibrium state. At the initial stage of informatization construction, the head office increased financial investment in informatization construction in factory 3 to support their work. The factory manager made plain the necessity of transformation from industrialization to informatization, discussed cooperation with informatization project suppliers, encouraged employees to actively participate in informatization construction, helped employees to realize that informatization can greatly improve work efficiency, urged the relevant responsible persons to cooperate in the development of information system projects, managed the information system projects well, and set the milestones for major projects. All kinds of efforts converged to significantly improve informatization performance. The good organizational atmosphere and training and learning channels in the later period effectively trained employees’ technical ability, corrected bad working habits, helped employees to master new working methods, improved work processes, and improved the overall quality of employees, which enabled them to develop in a balanced manner and be stable at points (3, 8, 30), meeting the expectations of the factory for information performance.

Comparing system (2) with the adjusted system (3), it is not difficult to find that the balance points (3, 8, 30) of system (5) are obviously superior to the former two, and the informatization performance can be significantly improved with relatively little investment in human and financial resources. At the same time, the good organizational environment significantly improved the skills of employees, which provides a good technical foundation for factory 3 to increase its informatization construction into the next equilibrium state. From this point of view, it is not a bad thing for the enterprise that system (4) is in a chaotic state. Instead, it gives the head office an early warning, helps managers to tap into new opportunities that may arise in the process of informatization construction in time, sets the expectation for informatization development, adjusts the timetable for strategic deployment, accelerates the transformation and upgrading of the factory, improves the informatization performance and technical ability of employees, and enhances the comprehensive competitiveness of the factory, thus providing major benefits to the head office.

## 4. Discussion

The research results of this study show that the dynamic evolution model of manufacturing enterprise informatization development is reasonable, and the chaotic linear controller designed is effective. Case studies verify that the competition and cooperation among top management support, employees’ technical ability, and information performance will make the evolutionary trajectory of the dynamic system completely different. Factories 1 and 2 entered an equilibrium state, while factory 3 entered a chaos state. The balance point of factory 1 met the expectations of the head office. The balance point of factory 2 was not satisfied, which inspired managers to adjust the development strategy in time according to the actual situation, so that the system entered a new evolutionary trajectory and finally stabilized at the next expected balance point. The chaotic state of the factory 3 is a vicious cycle of the “dead” state, and the three values will have exponential differences over time, which gives the head office an early warning. Managers should grasp the expected value of the future information development and re-deploy the strategy according to the dynamic system after linear control, making it stay away from the chaotic state and enter a new equilibrium state so that it can be stable at the expected value. Combining systems (2–5) with the adjusted system (3), it is not difficult to find that the dynamic system must converge if it wants to evolve toward the expected value. After reaching an expected value, it needs to adjust the strategy to enter the next evolutionary trajectory; otherwise, it will stagnate, and the next evolution may converge or diverge. The divergent one needs to be controlled to converge. If the stable point of convergence does not meet expectations, it still needs to be adjusted until it meets the expectations. In this respect, we know that the transformation from industrialization to informatization is a process of constant adjustment and continuous advancement.

The study has the following characteristics:(1).Model: According to the relevant literature research and the research data of IOII in the Pearl River Delta region of China, a mathematical model is established, and the linear controller is designed in combination with enterprise practice, which has a certain theoretical and practical basis; the modeling method is scientific.(2).Case analysis: To further analyze the evolutionary mechanism of the informatization in manufacturing enterprises and analyze the interaction among top management support, employees’ technical ability, and informatization performance, the parameters of the model are calculated based on the scores of experts in different factories of the case company, and the time series and equilibrium points of each dynamic system are analyzed with MATLAB tools. The parameters are adjusted for an equilibrium state that does not meet the organization’s expectations, in combination with management practice, the chaotic characteristics are identified by using the maximum Lyapunov exponent, and chaotic linear control is carried out for the special system. The selection of methods and the use of tools are scientific.

However, this study inevitably has some shortcomings. The informatization development of manufacturing enterprises is influenced by many factors both inside and outside the organization, such as government policies, organizational culture, and leader quality. This study cannot provide evidence of these aspects. In addition, the case analysis, although it can verify the interaction among them, can only reflect the information evolution of the three factories of the company but cannot fully represent their evolution in this industry or other industries. The model constructed has certain reference value, but the model parameters should be selected according to the actual situation. In other companies, there may be stable convergence, an unstable saddle point, periodic bifurcation, chaos, and/or a limit cycle. This situation will be explained and compared in detail in a future article.

## 5. Conclusions

In this study, a dynamic evolutionary model of top management support, employees’ technical ability, and the informatization performance of manufacturing enterprises was constructed, and the model parameters were obtained by case data to carry out simulation experiments. The chaotic system was identified for chaotic characteristics, and a linear controller was designed to eliminate the chaotic state and enter a stable state. The results show that a factory in which the dynamic system evolves into a stable state can meet the development needs of enterprises, but a factory that does not meet the needs can still give managers inspiration to fine-tune their strategy in a timely manner to make the system converge to a new equilibrium point. A factory that has evolved into chaos is in a bad situation, yet chaos can represent a new opportunity. Managers can control the chaos by re-deploying the factory information strategy; linear control can stabilize the system to the expected value and accelerate the development of information construction toward the expected state of enterprises.

The transition from industrialization to informatization is a process of continuous promotion and adjustment. Organizations should pay attention to the development of top management support and staff skills in strategic planning for the future. It is not always a good thing for the three to co-evolve into a stable state, and it is not always a bad thing to enter a chaotic state. Managers should adjust their strategies according to different situations and the actual development needs of enterprises, and grasp the process of transformation and upgrading, so as to ensure the success of informatization construction, promote deep IOII, guide enterprises onto a beneficial development path, and enhance their core competitiveness.

Admittedly, chaos is a complex problem. In the future, we will analyze the complexity characteristics of chaotic systems through other complex system theories and analyze its dynamic characteristics and possible new interesting problems in more detail, in combination with the periodic bifurcation of chaotic systems. In our future work, we will transform chaotic time series from the time domain to the frequency domain, based on fast Fourier transform (FFT), calculate its spectral entropy and complexity measure by the spectral entropy method (SE) and C_0 complexity algorithm, and then analyze and compare the richer dynamic characteristics of continuous chaotic systems by comparing the periodic bifurcation diagram, SE graph, and C_0 complexity curve.

## Figures and Tables

**Figure 1 entropy-23-00681-f001:**
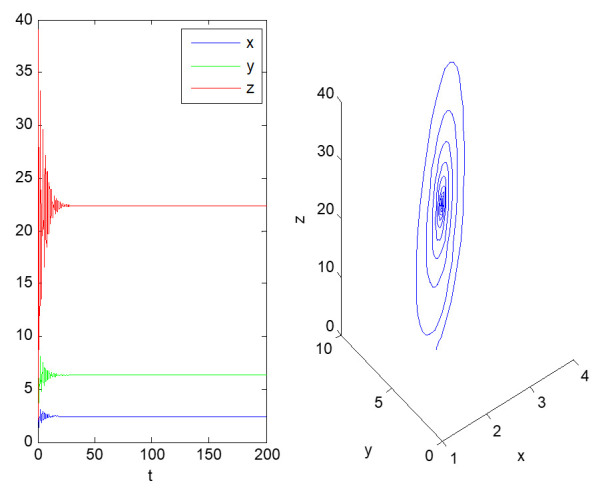
Time domain diagram (**left**) and three-dimensional phase diagram (**right**) of the evolution of dynamic system (2).

**Figure 2 entropy-23-00681-f002:**
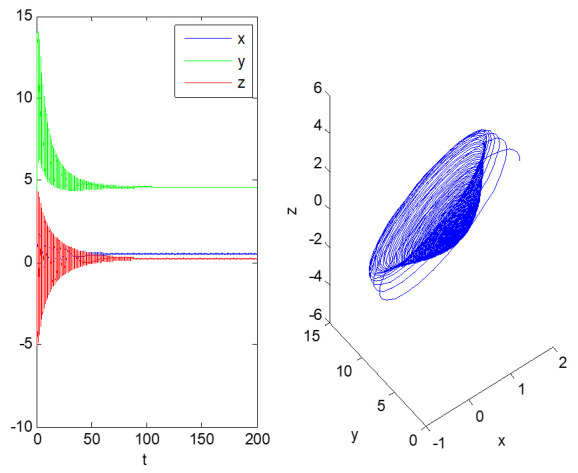
Time domain diagram (**left**) and three-dimensional phase diagram (**right**) of the evolution of dynamic system (3).

**Figure 3 entropy-23-00681-f003:**
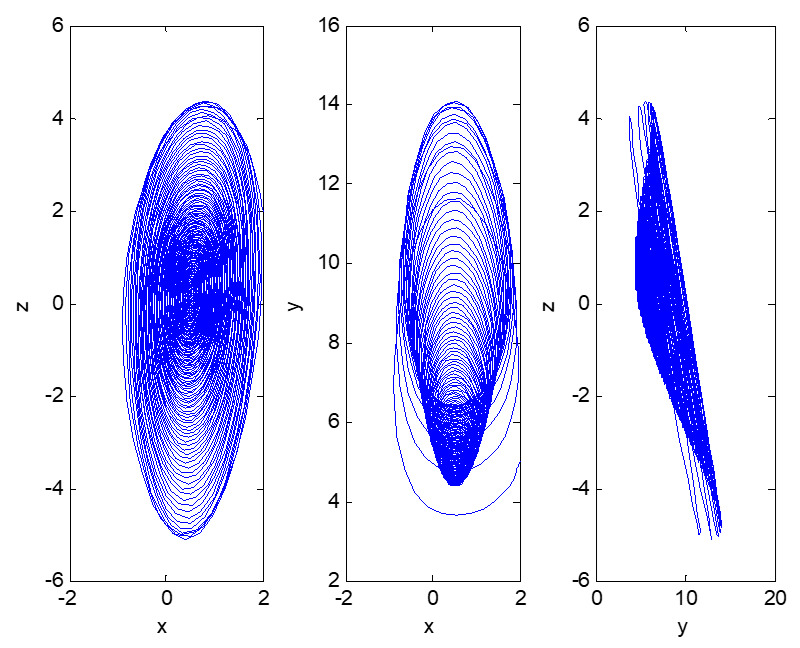
Two-dimensional phase diagram of the evolution of dynamic system (3).

**Figure 4 entropy-23-00681-f004:**
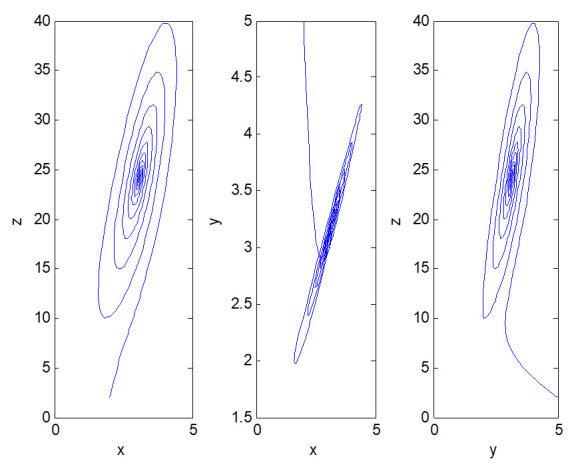
Two-dimensional phase diagram of the evolution of the adjusted dynamic system (3).

**Figure 5 entropy-23-00681-f005:**
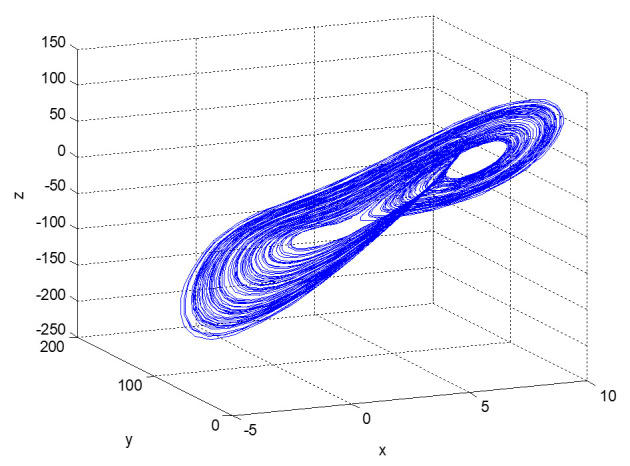
Three-dimensional phase diagram of the evolution of dynamic system (4).

**Figure 6 entropy-23-00681-f006:**
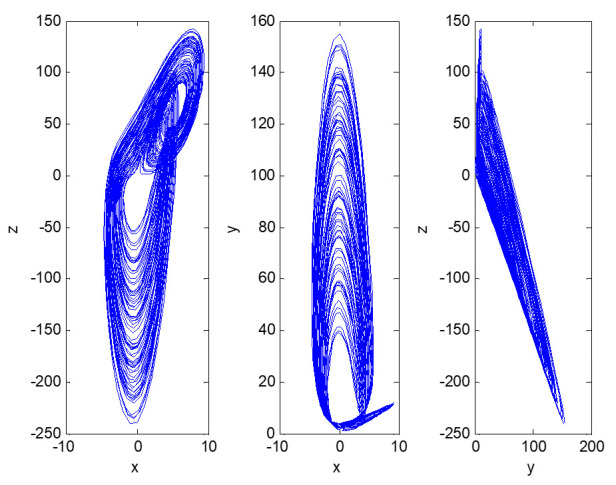
Two-dimensional phase diagram of the evolution of dynamic system (4).

**Figure 7 entropy-23-00681-f007:**
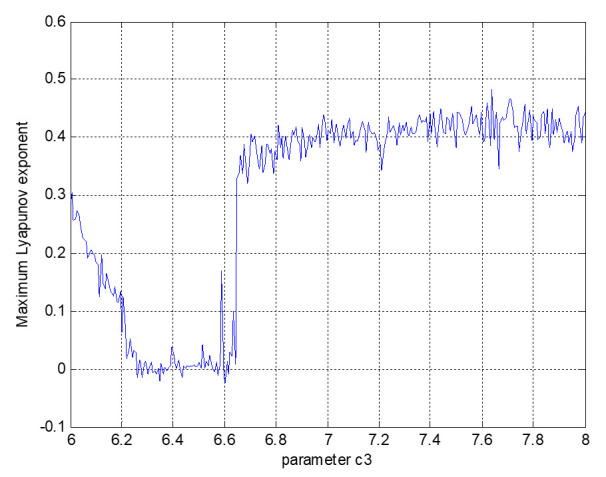
Maximum Lyapunov exponent of dynamic system (4), changing with c3.

**Figure 8 entropy-23-00681-f008:**
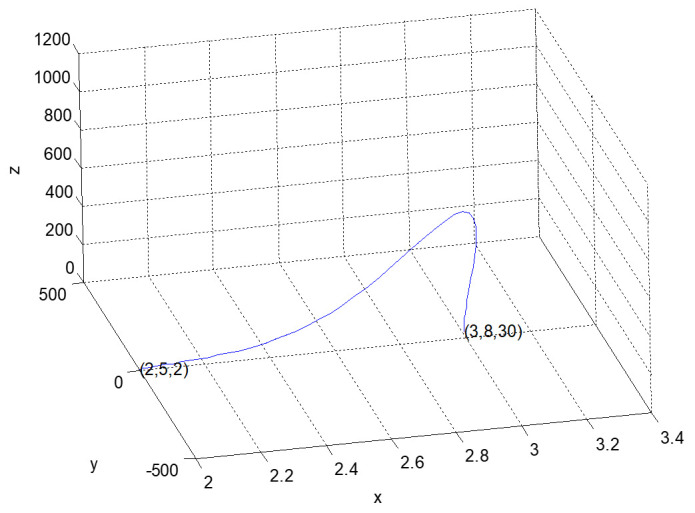
Evolutionary three-dimensional phase diagram of dynamic system (5).

**Table 1 entropy-23-00681-t001:** Conceptual description.

Concept	Content Description
Top management support	The top managers believe that informatization will bring great benefits to enterprises.
	The top managers strongly support informatization construction in terms of human resources and pay attention to talent introduction and skills training.
	The top managers support information construction in finance and actively introduce information software and hardware equipment.
	The top managers actively participate in the implementation of information projects.
	The top managers set specific goals and standards to supervise the progress of informatization.
	The top managers actively promote the application of informatization.
	The top managers actively attend various conferences and ceremonies of informatization.
	The top managers encourage learning from industry benchmarking enterprises and experts on issues related to informatization construction.
Technical ability of employees	Employees quickly adapt to information-based operations and their operational skills are improved, for example, via KPIs.
	Informatization has improved the management skills of middle-level employees, such as team ability and employee leadership.
	Informatization has become the main way for employees to work.
	The use of information technology has stimulated the professional and technological innovation ability of employees, such as intellectual property.
Informatization performance	The information system in the factory has unified basic data, and different systems can realize interconnection and intercommunication.
	Different departments can achieve collaborative work through information technology.
	Informatization has improved the operational efficiency of the factory.
	Informatization reduces the operating cost of the factory.
	Informatization promotes the smooth development of business processes.
	Informatization enhances the rapid response capability of the factory.
	Informatization promotes the improvement of management level.
	Informatization is conducive to making reasonable business decisions.
	Informatization has improved customer satisfaction.
	Informatization has improved the relationship with suppliers.

## Data Availability

Data sharing not applicable.

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
