# Peer review of "Dynamic Modeling and Chaos Control of Informatization Development in Manufacturing Enterprises"

_entropy, 2021, doi:10.3390/e23060681_

Round 1
Reviewer 1 Report
The article is devoted to the development of a mathematical model reflecting the dependence of production on management decisions. Despite the interesting ideas described in the manuscript, I cannot recommend it for publication due to the following reasons:
1. The article contains several broken links to figures.
2. It is not entirely clear in what quantities x, y are measured in formula (1)? How can top management support be measured? Could the Authors give an example of such a measurement in addition to expert judgment? Can the model include some more objective factors for assessing the employee's work and management decisions, for example, KPI?
3. Why is the model drawn this way? If the purpose of the study is to reveal the dependence of productivity on managerial decisions, then why are they not specified in the form of control laws in the control system?
4. Formulas (2) - (5) are modeled with constant parameters, while in reality these parameters will change.
5. What integration method was used to model the differential equations? As a rule, the method and sampling step quite strongly affect the accuracy of the resulting model.
6. In what units is T measured Figures 1,2? Years, months?
7. For the presented model it is not shown that it corresponds to reality. It would be visual to take data from the past, analyze management decisions and show that the model reproduces the current state of production. Unfortunately, this is not reflected in the article, so the relevance of the model is not confirmed.
Author Response
The article is devoted to the development of a mathematical model reflecting the dependence of production on management decisions. Despite the interesting ideas described in the manuscript, I cannot recommend it for publication due to the following reasons:
1.The article contains several broken links to figures.
Answer: We have re-explained all the figures.
- It is not entirely clear in what quantities x, y are measured in formula (1)? How can top management support be measured? Could the Authors give an example of such a measurement in addition to expert judgment? Can the model include some more objective factors for assessing the employee's work and management decisions, for example, KPI?
Answer: In formula 1, no specific values are set for x and y, but the corresponding contents in table 1 are used to measure each variable, and the expert scoring method is used to score the mutual influence between each variable. In the simulation, the initial values of x, y and z are set to analyze the changes of the three variables. In addition to the expert's judgment, we can also know the changes of x and y in different factories in the form of questionnaires, but the contents in Table 1 will be more specific. For example, through the number and structure of information technology talents, the number of information technology training courses and the number of employees attending classes in a period, we can measure that the top managers of the factory support informatization construction in human resources and pay attention to talent introduction and skills training. You can also analyze the changes of x and y from the factory reports or related records, such as the frequency of factory top managers participating in informatization project meetings and related ceremonies at key nodes in a period, as well as absentee records, to measure the top managers active attendance at various informatization meetings and ceremonies. Some objective indicators have been added to the technical ability of employees, as shown in Table 1.
- Why is the model drawn this way? If the purpose of the study is to reveal the dependence of productivity on managerial decisions, then why are they not specified in the form of control laws in the control system?
Answer: First, through visits and surveys of companies, we found that the shortage of professional talents and managers' lack of a clear understanding of informatization strategies are the main reasons that make China's manufacturing "big but not strong". Therefore, this article establishes the use of top management support, employee technical capabilities and informatization performance to explore the dynamics in the process of corporate informatization construction and provide a scientific guide for the allocation of corporate resources and the introduction and training of professional talents. Secondly, the research purpose of this article is to explore the influence process of stop management support and technology ability on the advancement of informatization in the process of informatization construction. Finally, the dynamic system studied in this paper is a flexible system, which is controlled in real time according to the development of enterprise informatization. Therefore, there is no way to specify them in the form of control laws in the control system.
4.Formulas (2) - (5) are modeled with constant parameters, while in reality these parameters will change.
Answer: Yes. These parameters in this article are not constants but are constant in a certain stage of development. If viewed from different stages of development, they will change. Please refer to section 3.1 in the text.
5.What integration method was used to model the differential equations? As a rule, the method and sampling step quite strongly affect the accuracy of the resulting model.
Answer: This article first establishes a system of differential equations, and then linearizes the system of equations to obtain a system of differential equations.
6.In what units is T measured Figures 1,2? Years, months?
Answer: According to the quarter of the enterprise, 3 months are usually taken as a unit.
7.For the presented model it is not shown that it corresponds to reality. It would be visual to take data from the past, analyze management decisions and show that the model reproduces the current state of production. Unfortunately, this is not reflected in the article, so the relevance of the model is not confirmed.
Answer: We have explained in the article that data is obtained by scoring by experts, and then based on the scoring records of experts to form a time series. Use the time series to establish the difference equations, and then the equations are continuous. Later, Matlab was used to analyze the evolution of the equations, and the construction level, dynamic process, and development trajectory of informatization under different development strategies were given.
Reviewer 2 Report
The manuscript is terribly verbose.
For a submission in a scientific journal, there are too many verbose disgressions, including some sentences that could be interpreted as close to propaganda, which does not have its place in a scientific journal. The Reader is submerged by a bunch of boring information.
What is the point in saying e.g. that "only 22.8% of 49 these enterprises have achieved integration improvement and innovation breakthrough, 50 and most of them are in the single coverage stage, accounting for 52.4%....."
Apart from this, the core of the paper is a trivial consideration of a non-linear coupled ode system with a computation of Lyapunov exponent.
There might be some interest in the conceptual consideration by the authors. The current presentation is just terrible.
I suggest that the authors drastically cut into their manuscript to extract the essentials, present well their manuscript and go further into the analysis of the results. From the purely mathematical side, there is nothing new here. So there needs to be some clear interest from the societal side for the paper to become acceptable.
Author Response
The manuscript is terribly verbose.
For a submission in a scientific journal, there are too many verbose disgressions, including some sentences that could be interpreted as close to propaganda, which does not have its place in a scientific journal. The Reader is submerged by a bunch of boring information.
What is the point in saying e.g. that "only 22.8% of 49 these enterprises have achieved integration improvement and innovation breakthrough, 50 and most of them are in the single coverage stage, accounting for 52.4%....."
Apart from this, the core of the paper is a trivial consideration of a non-linear coupled ode system with a computation of Lyapunov exponent.
There might be some interest in the conceptual consideration by the authors. The current presentation is just terrible.
I suggest that the authors drastically cut into their manuscript to extract the essentials, present well their manuscript and go further into the analysis of the results. From the purely mathematical side, there is nothing new here. So there needs to be some clear interest from the societal side for the paper to become acceptable.
Answer:Dear reviewer, thank you for your valuable comments. We have made a lot of changes based on your proposal, deleted all the redundant information, and then explained the social significance of this model in more detail.
Reviewer 3 Report
In the paper, a dynamical system is proposed to model the cooperative evolution among support, employees, and IOII. Where, according to an adequate selection of parameters, the system is stable to an expected value. The authors compute the Largest Lyapunov Exponent (LLE) to identify a chaotic behavior from time series. The Following are my comments:
1-This paper has a little fit to the scope of Entropy. There is no information related to chaos control, entropy, and complexity in the Introduction focus on manufacturing enterprises. Please improve your Introduction section by considering these concepts.
2-There is an error in line 182.
3-Table 1, is not referenced in the main text
4-Letters related to state variables should be written in mathematical mode, please see line 184+185
5-The article lacks a deep analysis of the dynamic system.
6-There is an error in lines 237-238.
7-Please add the algorithm to compute the Largest Lyapunov Exponent.
8- There are some “Error! Reference source not found”. This offers an impossibility for a correct read of the paper.
9-It is well known that the entropy emerges in chaotic systems a Spectral Entropy Analysis or C_0 Complexity Analysis needs to be included in the proposal related to a set of parameters. Finally, Improve the conclusion by considering the comments and give a clear future research direction.
Author Response
In the paper, a dynamical system is proposed to model the cooperative evolution among support, employees, and IOII. Where, according to an adequate selection of parameters, the system is stable to an expected value. The authors compute the Largest Lyapunov Exponent (LLE) to identify a chaotic behavior from time series. The Following are my comments:
1.This paper has a little fit to the scope of Entropy. There is no information related to chaos control, entropy, and complexity in the Introduction focus on manufacturing enterprises. Please improve your Introduction section by considering these concepts.
Answer: We have re-explained the introduction and pointed out that the informatization construction is a complex and dynamic process with chaos.
2.There is an error in line 182.
Answer: We have revised it again, please refer to line 192.
3.Table 1, is not referenced in the main text.
Answer: We have already quoted, please refer to line 219
4.Letters related to state variables should be written in mathematical mode, please see line 184+185.
Answer: We have used the math formula editor to rewrite it.
5.The article lacks a deep analysis of the dynamic system.
Answer: In the follow-up case analysis, we conducted further analysis to illustrate the dynamics and stability of the system evolution.
6.There is an error in lines 237-238.
Answer: We have made changes, please refer to lines 247-249.
7.Please add the algorithm to compute the Largest Lyapunov Exponent.
Answer: We have added an algorithm, please refer to line 367-382
8.There are some “Error! Reference source not found”. This offers an impossibility for a correct read of the paper.
Answer: We reviewed the literature again, and now the literature can be searched online.
9.It is well known that the entropy emerges in chaotic systems a Spectral Entropy Analysis or C_0 Complexity Analysis needs to be included in the proposal related to a set of parameters. Finally, Improve the conclusion by considering the comments and give a clear future research direction.
Answer: The suggestion has been adopted, and we have added the future research direction of the conclusion part according to your opinions.
Round 2
Reviewer 1 Report
Dear Authors,
Despite the significant revision of the manuscript, I still cannot recommend this article for publication.
1. Since the article presents a model of a chaotic system, which is subsequently linearized, it is not entirely clear why the title says "chaos control." Сhaos cannot arise in linearized models. I consider this to be a gross violation. Moreover, the reliability of the obtained numerical results raises questions since linearization significantly affects the properties of the discrete model.
2. The study of the bifurcation properties of the system depending on the values of the parameters is one of the main components of any study of nonlinear systems. This is because a sudden change in a parameter can lead to unexpected system behavior. And this behavior seems to be worth controlling, as the title says. However, the analysis of the system's behavior has not yet been carried out.
3. No confirmation of the relevance of the model, all the more considering that it is linearized.
Therefore, I still insist on the rejection of this manuscript.
Reviewer 2 Report
The manuscript remains verbose yet some digressions have been suppressed which improves the reading.
This is a mixture of econmics, social sciences and notions of chaos theory that could have some interest and I therefore recommend the publication of the revised version of the manuscript.
Reviewer 3 Report
The authors have improved the paper by considering the comments of the reviewer. I consider that the paper fits into the journal's scope and now can be accepted